

# Managed aquifer recharge with reverse-osmosis desalinated seawater: modeling the spreading in groundwater using stable water isotopes

Yonatan Ganot[1,2], Ran Holtzman[2], Noam Weisbrod[3], Anat Bernstein[3], Hagar Siebner[3], Yoram Katz[4], and Daniel Kurtzman[1]

[1]Institute of Soil, Water and Environmental Sciences, The Volcani Center, Agricultural Research Organization, Rishon LeZion, 7505101, Israel
[2]Hydrology and Water Resources, The Hebrew University of Jerusalem, Rehovot, 7610001, Israel
[3]Department of Environmental Hydrology & Microbiology, Zuckerberg Institute for Water Research, Jacob Blaustein Institutes for Desert Research, Ben-Gurion University of the Negev, Midreshet Ben-Gurion, 8499000, Israel
[4]Mekorot, Water Company LtD, Tel Aviv, 6713402, Israel

*Correspondence to*: Yonatan Ganot (yonatan.ganot@mail.huji.ac.il)

**Abstract.** The spreading of reverse-osmosis desalinated seawater (DSW) in the Israeli Coastal Aquifer was studied using groundwater modeling and stable water isotopes as tracers. The DSW produced at the Hadera seawater reverse osmosis (SWRO) desalination plant is recharged into the aquifer through infiltration pond at the managed aquifer recharge (MAR) site of Menashe, Israel. The distinct difference in isotope composition between DSW ($\delta^{18}$O=+1.41; $\delta^2$H=+11.34‰) and the natural groundwater ($\delta^{18}$O=–4.48 to –5.43‰; $\delta^2$H=–18.41 to –22.68‰) makes the water isotopes a preferable tracer compared to widely-used chemical tracers, such as chloride. Moreover, this distinct difference can be used to simplify the system to a binary mixture of two end members: desalinated seawater and groundwater. This approach is especially robust when spatial data of stable water isotopes in the aquifer is scarce. A calibrated groundwater flow and transport model was used to predict the DSW plume distribution in the aquifer after 50 years of MAR with DSW. The results show that after 50 years 94% of the recharged DSW was recovered by the production wells at the Menashe MAR site. The presented methodology is useful for predicting the distribution of reverse-osmosis desalinated seawater in various downstream groundwater systems.

## 1 Introduction

Desalinated seawater global production is projected to double by 2040 while extending its geographical extent (Hanasaki et al., 2016). In some regions, desalinated seawater (DSW) is already the main source for fresh water (Dawoud, 2005). In Israel, for example, DSW reached 80% of the domestic and industrial fresh water supply in 2017 (Israel Water Authority, 2017). This growing use of DSW affects downstream water systems such as reservoirs (Ronen-Eliraz et al., 2017; Negev et al., 2017; Stuyfzand et al., 2017; Ganot et al., 2017, 2018), wastewater treatment plants (Lahav et al., 2010; Negev et al., 2017) and agricultural irrigation (Lahav et al., 2010; Yermiyahu et al., 2007). One direct way by which DSW use affects the





water budget is Managed Aquifer Recharge (MAR). MAR using different water sources has been practiced for over 5 decades as part of the integrated water resource management of Israel (Dreizin et al., 2008; Gvirtzman, 2002), and is becoming a major component of water management in many Mediterranean countries (Rodríguez-Escales et al., 2018).

Excess DSW produced in Israel due to operational constraints made it an attractive alternative source for MAR, raising the need to understand its effect. While the relatively rapid hydrological and geochemical processes (timescales of hours to weeks) of this new MAR activity were recently monitored and modeled (Ganot et al., 2017, 2018; Ronen-Eliraz et al., 2017), the potential long-term (months to decades) impact of this process on the natural aquifer is yet unknown, lacking observations and quantitative studies.

Stable water isotopes $^{18}$O and $^{2}$H are excellent tracers for water generated by seawater reverse osmosis (SWRO) desalination. The lack of fractionation during the reverse-osmosis process, in contrast with various isotope-fractionation processes occurring in natural fresh water (Al-Basheer et al., 2017; Gat, 1996; Kloppmann et al., 2008a, 2008b), is the cause of the distinct difference in isotope composition between reverse-osmosis DSW and groundwater originating from natural fresh water (Ganot et al., 2018; Kloppmann et al., 2018; Negev et al., 2017). For example, the advantage of using $^{18}$O and $^{2}$H as a

quantitative tool for tracing DSW mixing with groundwater (GW) was recently demonstrated by comparing mixing ratios of chloride, carbamazepine and water isotopes in the soil-aquifer-treatment (SAT) site at the Shafdan MAR system, Israel (Negev et al., 2017).

Here, we use stable water isotope to trace spreading of DSW in the aquifer and the production wells within the MAR site of Menashe, Israel. The DSW are produced at the Hadera SWRO desalination plant, which operates since 2010 with an annual

production capacity of 130 million cubic meters (MCM). It is one of five large SWRO desalination plants (production capacity ≥ 90 MCM, per year per plant) that were built along the Mediterranean coast of Israel during 2005–2015 (Stanhill et al., 2015). The DSW is regularly supplied directly to consumers through the centralized national water system. Periodically, operational constraints such as maintenance of the national system prohibit distribution of the DSW; limited reservoir capacity makes storage of this expensive surplus of DSW in the aquifer through MAR operations the only feasible solution

(Ganot et al., 2017, 2018; Ronen-Eliraz et al., 2017).

Predicting the long-term DSW distribution in the aquifer and the production wells is the main objective of this study. We incorporate water isotope data of $^{18}$O and $^{2}$H in a regional GW flow and transport model (e.g., Boronina et al., 2005; Krabbenhoft et al., 1990; Liu et al., 2014; Reynolds and Marimuthu, 2007; Stichler et al., 2008) in order to predict DSW distribution in the aquifer. While the methodology for measuring the present mixing of DSW and GW was reported

previously (Negev et al., 2017), in the current study our GW modeling approach allows us to predict future mixing trends in the production wells of the Menashe MAR site. Predicting DSW distribution in the aquifer is of main interest from water quantity (estimating the recovery potential of DSW originate from MAR) and quality perspectives (e.g., Birnhack et al., 2011; Ganot et al., 2018 and references therein).





## 2 Methods

**2.1 Study area**

The Menashe MAR site is located on sand dunes 28 m above sea level, in the northern part of the Israeli Coastal Aquifer, an unconfined sandy aquifer stretching over an area of 2000 km$^2$ along the Mediterranean coast (Fig. 1a). The local climate is Mediterranean, with annual average temperature of 20.2°C, and annual mean precipitation of 566 mm/yr (Israel Meteorological Service, 2014). The aquifer thickness varies from 100 m on the coastline (to the west of the Menashe site) to

few meters in the east. It is composed of Pleistocene calcareous sandstone interleaved with discontinuous marine and continental silt, and clay lenses. Thick Neogene clay (Saqiye Group), which is highly impermeable, underlies the aquifer (Kurtzman et al., 2012). Regional groundwater level is ~3 m above mean sea level ( September 2014, Israel Water Authority, 2014) and the characteristic aquifer properties are: hydraulic conductivity of 10 m/d, storativity of 0.25 and porosity of 0.4 (Shavit and Furman, 2001).

The Menashe MAR site diverts the natural ephemeral flows from the Menashe-Hills streams into a settling pond and from there to three infiltration ponds. Production wells that encircle the site recover the recharged water from the aquifer (Sellinger and Aberbach, 1973). In the last few years, the southern infiltration-pond is dedicated for infiltration of surplus of DSW from the Hadera SWRO desalination plant, located 4 km to the west, on the coastline (Fig. 1b).

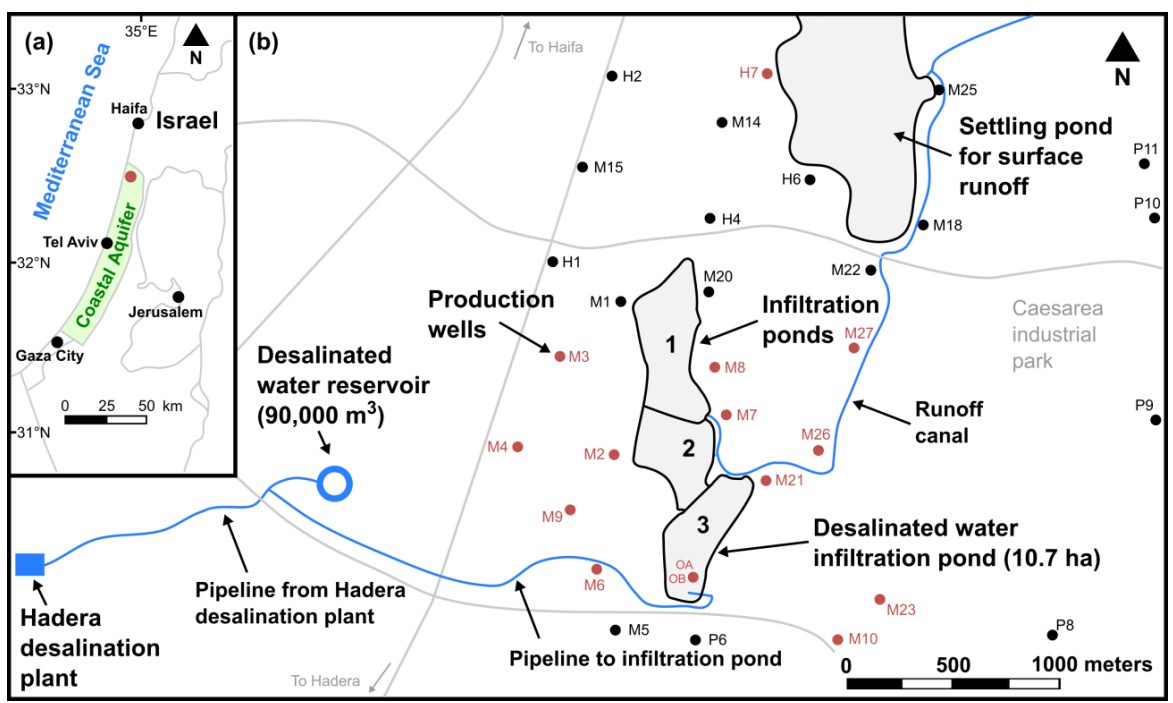

**Figure 1. Map of the study area. (a) Location of the Israeli Coastal Aquifer and the Menashe MAR site (red circle). (b) The Menashe MAR site. Surplus of desalinated seawater is delivered from Hadera SWRO desalination plant (lower left) to the southern infiltration basin (pond 3). The red dots represent wells that were sampled for water isotope analysis.**



## 2.2 Water sampling

Groundwater from 14 wells at the Menashe MAR site were sampled biannually during 2015 to 2017. In addition, DSW were sampled from the infiltration pond inlet pipe during MAR events. Stable water isotopes (expressed as $\delta^{18}O$ and $\delta^2H$ in ‰ vs. the VSMOW – Vienna Standard Mean Ocean Water) were measured by Cavity Ring-Down Spectroscopy (CRDS) analyzer (L2130-i, Picarro).

## 2.3 Groundwater flow and transport model

A detailed three-dimensional transient water flow and solute transport model was set up in order to estimate DSW spreading in the aquifer at the Menashe site area. The model covers an area of 65 km$^2$ including a western out-shore strip of 9 km$^2$ (Fig. 2a). The geological data processed from well logs, geological and structural maps served as the basis for the conceptual model, constructed via the GMS software package (version 10.3; www.aquaveo.com). The variety of rock types was grouped into four hydro-geological units, each characterized by a set of hydrological properties (Table 1). Over 100 well

logs were analyzed using the T-PROGS software (Carle, 1999) and provided the spatial distribution of the hydro-geological units. This geo-statistically generated unit array, conditioned to the boreholes logs, was combined with structural map data of the major marine clay lenses present in the aquifer. The resulting model hence reflects the hydro-geological units' proportions and transition trends as well as the division into sub aquifers by marine clay within the western part of the aquifer (Fig. 2b, c).

The model domain was discretized horizontally into 70 X 70 m mesh cells. The vertical section of the aquifer, of thickness ranging 50–100 m from east to west, was divided into 24 layers with vertical spatial-resolution of 5 m or smaller. The model bottom boundary was defined by the impermeable Saqiye Group underlying the aquifer. The model top boundary was defined by the water table representing an unconfined aquifer. Boundary conditions along the northern, eastern and southern model boundaries were set to be of transient head, based on periodical water level measurements. The western boundary was

set to a constant head boundary dictated by the sea level. Initial conditions were based on static heads measured at several dozens of production and observation wells included in the model. Sources and sinks in the flow model include recharge by precipitation, MAR (both runoff and DSW recharge) and production wells. Natural recharge from precipitation was based on adjacent rain gauge measurements (Gan Shemuel) using an average recharge coefficient of 0.4 (which is representative of sands). Recharge flux of DSW by MAR activity was calculated by a variably-saturated model of the upper 30 m of the

sediment under the southern infiltration pond (Ganot et al., 2017). Pumping activity of the production wells was based on a database from the national water company of Israel, Mekorot.





**Figure 2. The model used in simulations of water flow and solute transport. (a) The modeled area and boundary conditions. (b) The major continuous marine clay lenses, and the boreholes log. (c) The combined deterministic and geostatistically-generated material array representing the aquifer in the model.**



**Table 1. Major rock types in the study area grouped into four hydro-geological units**

| Hydro-geological unit | 1 | 2 | 3 | 4 |
|---|---|---|---|---|
| **Rock types** | Gravel, beach rock, Kurkar with shells/gravel | Calcareous sandstone (Kurkar), sand | Loam, sandy loam, loamy sand, marine silty sand | Clay/silt of marine or terrestrial origin |
| **Hydraulic conductivity (K)** | High | Medium | Low | Very low |
| **Unit proportions (%)** | 4 | 59 | 23.5 | 13.5 |

The transport model considers the stable water isotopes $^{18}O$ and $^{2}H$ as conservative tracers, i.e. neglecting isotope fractionation. We normalize the tracer concentration as $C=(\delta - \delta_{min})/(\delta_{max} - \delta_{min})$, where $\delta$ is the isotope composition of $\delta^{18}O$

or $\delta^{2}H$ in the aquifer, and $\delta_{min}$ and $\delta_{max}$ the minimum and maximum isotope composition. Since practically $\delta_{max}= \delta_{DSW}$, the normalized concentration of DSW is $C_{DSW}=1$, whereas that of GW ranges from $C_{GW}=0$ ($\delta^{18}O=–5.43‰$, $\delta^{2}H=–22.68‰$) to $C_{GW}=0.13$ ($\delta^{18}O=–4.48‰$, $\delta^{2}H=–18.41‰$). Boundary conditions of the transport model are of specified mass flux ($=qC$, where q is the specific discharge), with zero flux at the bottom boundary (considered impermeable), as well as zero flux at the northern, eastern and southern boundaries, and also with the precipitation and the runoff-ponds source terms due to their

GW isotopes composition ($C_{GW}=0$). Mass flux with DSW isotopes composition ($C_{DSW}=1$) is given at the western boundary (sea) and the DSW infiltration pond source term. The validity of the use of a single value ($C_{GW}=0$) for the GW mass-flux boundaries, in light of the range of isotope composition in the aquifer prior to MAR of DSW ($\delta^{18}O=–4.48$ to $–5.43‰$ and $\delta^{2}H=–18.41$ to $–22.68‰$), is discussed in Section 3.2.3. Initial conditions were set by interpolating the water isotope data from several production wells.

The MODFLOW (Harbaugh et al., 2000) and MT3DMS (Zheng and Wang, 1999) codes were used through the GMS user interface to solve numerically the flow and transport models, respectively. Both codes, which use finite difference scheme, are considered reliable and are therefore widely used for regional aquifer modeling (Zhou and Li, 2011). The flow and transport model was calibrated using a dataset from 2015 to 2017. During 2015, 2016 and 2017 a volume of 2.6, 1.3 and 0.6 MCM of DSW were recharged, respectively, at the MAR Menashe site. In these years the MAR events were non-continuous

discharge of DSW to pond 3 (Fig. 1b) during January and/or February (Ganot et al., 2017, 2018). In addition, a volume of 3.2 and 1.6 MCM of runoff water were discharged to the settling pond during 2015 and 2017, respectively.

## 3. Results and discussion

### 3.1 Water isotopes

The distinct difference between the water isotopes of the production wells and DSW is shown in a $\delta^{2}H$ vs. $\delta^{18}O$ diagram for

the period of 2015 to 2017 (Fig. 3a). During 2016 and more prominently in 2017, few wells show a progressive change in composition towards higher isotope values – a transition from GW towards DSW on the mixing line (Fig. 3a), which





indicates mixing with DSW, while most wells retain constant isotope composition. Note that for all samples in Fig. 3a there is a strong linear correlation between $\delta^{18}O$ and $\delta^2H$ ($R^2$=0.9991); thus, hereafter we only report $\delta^2H$ as a tracer.

The isotope composition of $\delta^2H$ and the concentration of chloride are shown for comparison in nine wells during the years 2010 to 2018 (Fig. 3b). The chloride concentration of DSW at the Menashe MAR site is always lower than 10 mg/l (Ganot et al., 2018), while in the local GW it is found in a wider range of 40 to 140 mg/l. The large chloride concentration variability in the different wells prior to MAR with DSW (before 2015) suggests that various water sources feed the aquifer. Moreover, the breakthrough of DSW in wells M2, M6 and M9 captured by an increase in $\delta^2H$ is not reflected in the chloride concentration (expected to decrease). This implies that chloride – in general a widely used conservative tracer, is less sensitive to reverse-osmosis DSW in natural fresh GW systems and therefore less useful for its detection.

Finally, we note that the very different DSW signature in terms of $\delta^2H$ from the other water sources in the Menashe site reduces the problem of mixing various water sources to a binary system: (i) DSW and (ii) all other natural sources. This is because the signatures of runoff water ($\delta^{18}O$=–4.77‰ and $\delta^2H$=–19.5 ‰) and rainwater ($\delta^{18}O$=–5.8‰ and $\delta^2H$=–19.9‰; Gat and Dansgaard, 1972) are very similar to that of the local GW.

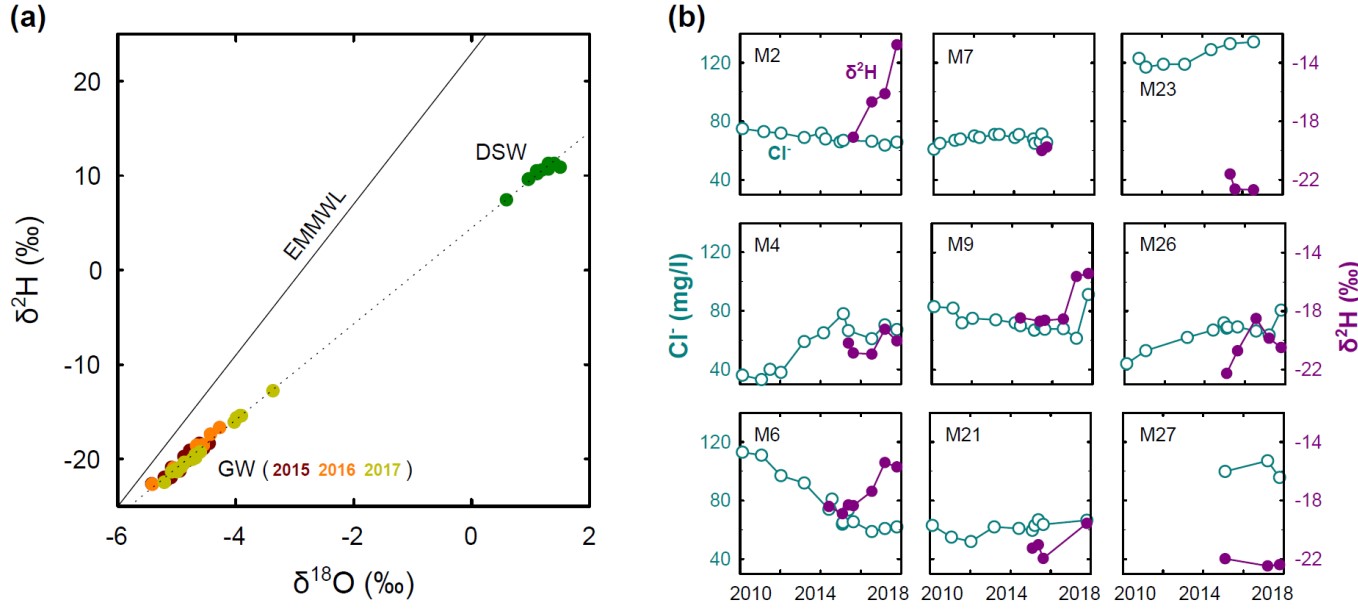

**Figure 3.** (a) Water isotopic composition of the production wells (GW) and reverse-osmosis desalinated seawater (DSW); the eastern Mediterranean meteoric water line (EMMWL) is shown for comparison (Gat and Dansgaard, 1972). (b) Chloride (Cl⁻) and $\delta^2H$ sampled in nine production wells at the Menashe MAR site.

### 3.2 Model

#### 3.2.1 Calibration

The flow model was calibrated against head data from 13 wells (Fig. 4a). We used mainly continuous head data measured at two production wells, M5 and M8 (Fig. 4b). Well M5, situated 400 m SE of pond 3 and exploiting aquifer layers bounded


between −16 to −54 m MSL, was inactive during 2015–7, making it ideal for head monitoring. Well M8, situated 1 km north of pond 3 and exploiting aquifer layers bounded between −14 to −48 m MSL, was used for production during some of the

study period, and thus only selected head data (representing quasi-static heads) were used for calibration.

The transport model was calibrated against isotope data from 12 wells (M2–4, M6–10, M21, M23, and M26–27; Fig. 4c). Specifically, we used data corresponding to the breakthrough of DSW in the down-gradient (western) production wells near the DSW infiltration pond (M2, M6, and M9), since other wells showed smaller $\delta^2$H variations (Fig. 4d). The simulated groundwater heads and $\delta^2$H for the calibration period are generally in good agreement with observations. The calibrated

hydrological parameters are specified in Table 2.

**Figure 4. Model calibration.** (a) Comparison of simulated and observed hydraulic head. (b) Temporal variations of simulated and observed hydraulic head in wells M5 and M8. (c) Comparison of simulated and observed $\delta^2$H. (d) Temporal variations of simulated and observed $\delta^2$H in nine selected wells.





**Table 2. Calibrated parameters used for the different hydro-geological units.**

| Hydro-geological unit | 1 | 2 | 3 | 4 |
|---|---|---|---|---|
| Horizontal K (m d$^{-1}$) | 50 | 12 | 6 | 0.01 |
| Vertical K (m d$^{-1}$) | 12.5 | 3 | 1.5 | 0.01 |
| Specific storage (m$^{-1}$) | 0.002 | 0.0015 | 0.001 | 0.001 |
| Specific yield | 0.35 | 0.12 | 0.12 | 0.1 |
| Longitudinal dispersivity[a] (m) | 20 | 20 | 20 | 20 |
| Porosity | 0.35 | 0.19 | 0.17 | 0.1 |

[a]Transverse horizontal and vertical dispersivities are 0.1 and 0.01, respectively, of the longitudinal dispersivity (Burnett and Frind, 1987).

### 3.2.2 DSW spreading in the aquifer

Our simulations show that at the end of 2017 the DSW plume is spreading westwards (in the direction of the natural
hydraulic gradient, as expected), approaching the closest western production wells (M2, M6, M9; Fig. 5a). Note that the
production wells to the east (up-gradient) show constant $\delta^2H$, indicating no interaction with the DSW recharge. Variability of
$\delta^2H$ along the production wells screens (in the vertical direction), implies that the measured $\delta^2H$ is a mixture of several
aquifer layers (Fig. 5b).

The $\delta^2H$ variations shown in Fig. 5a reflect the DSW spreading in the aquifer. The highest $\delta^2H$ that was measured in the
aquifer (prior to MAR of DSW) was $\delta^2H=-18.41‰$ and therefore any value above it indicates mixing with DSW. However,
because the initial measured $\delta^2H$ values in the aquifer are in the range of $\delta^2H=-18.41‰$ to $-22.68‰$, the extent of DSW
mixing in each well is relative to its specific initial $\delta^2H$. This can be calculated by a mixing ratio (MR) approach with
$MR=(\delta_w - \delta_i)/(\delta_{DSW} - \delta_i)$, where $\delta_w$ is the $\delta^2H$ in the well, $\delta_i$ is the initial (background) $\delta^2H$ in the well and $\delta_{DSW}$ is the $\delta^2H$ of
DSW. The MR value of 0 and 1, implies original aquifer water and pure DSW, respectively. Fig. 5c shows the MR
(expressed in %DSW) of three down-gradient wells (M2, M4 and M6), two up-gradient well (M23 and M26) and an
observation well (OA) inside the DSW pond. Wells M2 and M6 have up to 20% DSW portion while M23 and OA retain
original aquifer water and almost pure DSW, respectively. At the end of 2017, about 7% of the recharged DSW was
recovered by the production wells.

Knowing the water composition of the aquifer and of DSW, and assuming a conservative transport of all the major ions, one
can estimate the water composition in a specific well based on the calculated mixing ratio, $[X]_w=MR\times[X]_{DSW}+(1-MR)\times[X]_i$.
Here $[X]_w$ is the (calculated) ion concentration in the well, $[X]_{DSW}$ is the ion concentration in the DSW, and $[X]_i$ is the initial
ion concentration (background) in the well. Diversion of the observed concentration from the calculated concentration can
give insight to the sediment-water reaction (e.g., Ganot et al., 2018; Ronen-Eliraz et al., 2017; Stuyfzand et al., 2017).



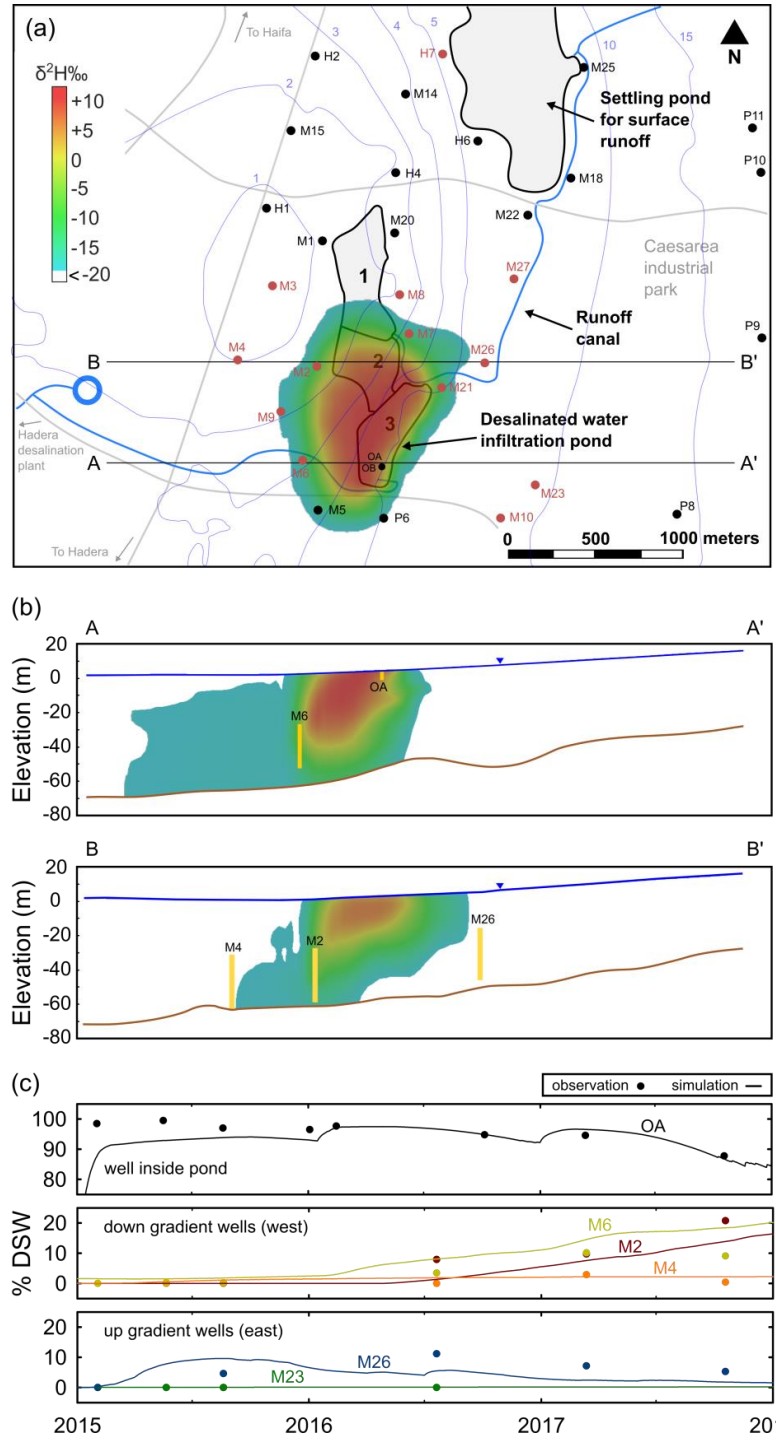

Figure 5. Simulation results showing DSW spreading at the end of 2017. (a) Plan view; colored area shows the DSW plume, white area indicates natural GW ($\delta^2$H<–20‰) and blue contours are GW head. (b) Cross-sections east-west through wells OA and M6 (A-A') and wells M4, M2 and M26 (B-B'); well screens are shown in yellow. (c) Observed and simulated DSW fraction (%) in selected wells along the cross-sections A-A' and B-B'.





### 3.2.3 Binary system assumption

The model was based on the assumption that all water types in this system can be described by two end-members sorted by their isotope composition: (1) the 'heavy' DSW ($\delta^2$H=+11.34‰); and (2) the 'light' natural water ($\delta^2$H=–22.68‰) which includes all other water types (rain, runoff and GW). As pointed out before, while DSW isotope composition is constant, that of the local natural water is more variable. To examine the validity of the assumption of binary $\delta^2$H values, we ran the simulation again for the same period of 2015 to 2017, but this time with the maximum value of GW $\delta^2$H=–18.41‰ (in all

GW boundaries and also as rain and runoff source) in order to check the model sensitivity to the natural GW isotope variability. We subtracted the isotope composition results of the two simulations in all model cells to produce an error map (Fig. 6a) of $\delta^2$H differences (Δ‰). In terms of $\delta^2$H composition in the production wells (Fig. 6b), the results of both simulations were similar (Δ‰<1), while some differences (up to Δ‰=4.3) were found in the domain boundaries and at the upper layer that was affected by rain and runoff recharge. Specifically, a notable difference is seen in the runoff settling pond

which is a source of natural water recharge. Nevertheless, for the area surrounding the DSW infiltration basin (pond 3), the binary assumption is valid due to the following conditions: (1) the distinct difference between the isotope composition of DSW and GW; (2) the model boundaries are relatively far (>2 km) from the source of MAR with DSW; and (3) the screens of the production wells are relatively deep (depth >50 m).  A major advantage of this assumption is that it allows to estimate mixing when the spatial data of water isotope is limited. This was exploited in the current study, where isotope data of the

model boundaries was unavailable.

### 3.2.4 Predicting long-term DSW spreading in the aquifer (2015-2065)

We test the extent of DSW spreading in the aquifer by performing long-term (50 years) simulation of MAR with DSW, considering 50 repeated annual cycles of the hydraulic conditions recorded in 2015, with a MAR event of 2.6 MCM (Fig. 7a). As expected, the water in the down-gradient (westwards) wells closest to the DSW pond, M2 and M6, are fully

exchanged by DSW after 10 years of MAR, while the up-gradient wells show little (M26) or no mixing (M23) with DSW (Fig. 7b,c). Interestingly, well M4 located further to the west, reaches a steady DSW mixing of almost 70% after about 35 years of MAR without being fully exchanged by DSW, while the DSW plume continues to progress further west. By the end of 2065, the total DSW volume of 130 MCM recharged at the infiltration pond is distributed as follows: 114 MCM (88%) is recovered by the western pumping wells (M2–9, P6), 8.4 MCM (6%) by the eastern pumping wells (M21, M26), with only

7.5 MCM (6%) remaining in the aquifer.

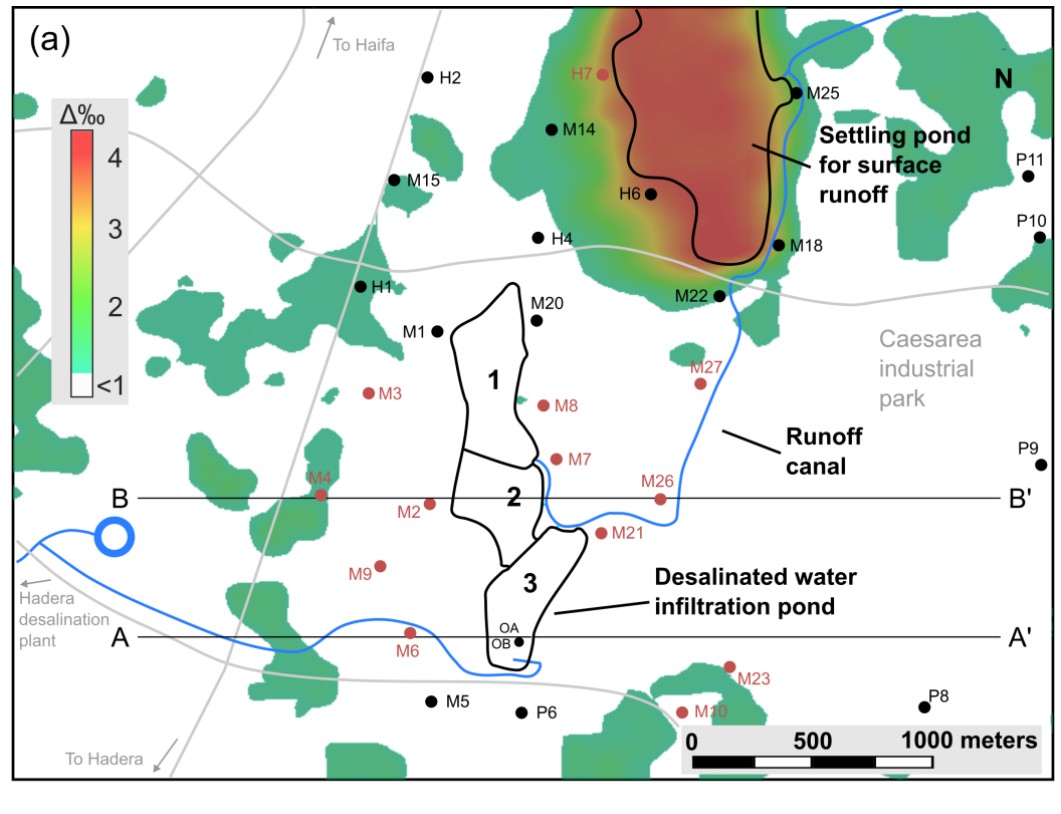

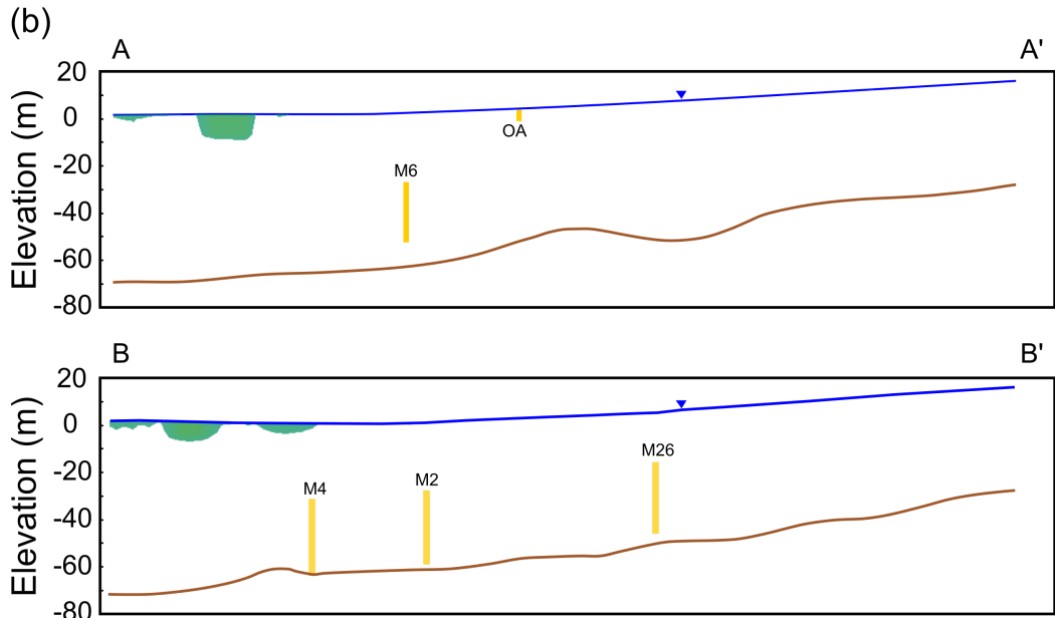

**Figure 6. Examination of the validity of the assumption of binary isotopic mixing. (a) Plan view of $\delta^2H$ difference ($\Delta‰$) between simulation results (2015–2017) with $\delta^2H_{max}=-18.41‰$ ($C_{GW}=0.13$) and $\delta^2H_{min}=-22.68‰$ ($C_{GW}=0$) at the end of 2017; white area indicates $\Delta‰<1$. (b) Cross-sections east-west through wells OA and M6 (A-A') and wells M4, M2 and M26 (B-B'); well screens are shown in yellow.**





**Figure 7. Long-term simulations of DSW spreading at the end of 2065 after 50 years of MAR. (a) Plan view; colored area shows the DSW plume, white area indicates natural GW (δ²H<−20‰) and blue contours are GW head. (b) Cross-sections east-west through wells OA and M6 (A-A') and wells M4, M2 and M26 (B-B'); well screens are shown in yellow. (c) Simulated DSW fraction (%) in selected wells along the cross-sections A-A' and B-B'.**






## 4. Conclusions

We track the fate of reverse-osmosis DSW that were introduced to groundwater by MAR, using stable water isotopes. The use of the water isotopes of $^{18}$O and $^{2}$H is advantageous in this system for two reasons: (1) there is a distinct difference between isotope composition of DSW and natural water; and (2) the water isotope composition of all natural water sources –
groundwater (GW), rain and runoff – is very similar. The former makes water stable isotopes a more sensitive tracer (compared to other natural conservative tracers such as chloride), whereas the latter reduces the problem to a binary mixture of two end-members: reverse-osmosis DSW and natural GW. We formulate a detailed three-dimensional GW flow and transport model, exploiting these advantages. The model, calibrated using field data (measured during 2015–2017), is used to predict the spreading of DSW in the aquifer during 50 years of MAR with DSW. Our simulation results show that most of
the recharged DSW (94%) is recovered by the production wells, indicating the efficacy of the Menashe MAR site. The presented methodology is valuable for predicting DSW plume spreading in natural groundwater aquifers.

### Acknowledgments

The research leading to these results received funding from the Germany-Israel binational scientific cooperation BMBF-MOST, Project WT1401. We thank Amos Russak and Raz Studny (Ben-Gurion University) for sampling and analysis
assistance.

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
