# Peer review of "Managed aquifer recharge with reverse-osmosis desalinated"

_Hydrology and Earth System Sciences, 2018_

## Referee Comment (RC1) · A. Furman (Referee) · 11 Jul 2018

This manuscript describes a case where water of unique quality recharges a local aquifer. The manuscript looks experimentally and through MODFLOW modeling at the expansion and mixing of the new water in the aquifer for a specific setup where recovery wells are located around the recharge pond. The concern here is desalinated water, and the main tool to distinguish these water from the background is stable isotopes.

[Figure]

Sincerely, I am not sure how to treat this manuscript. On one hand addresses a question of increasing concern (globally, not only in Israel), and it describes a work properly conducted, including a very nice combination of isotopic work and a flow and transport model. On the other hand, I'm asking myself at the end of the day "what's new?" Is there something that can really be learned from this manuscript? Is it describing something that was not clear from the beginning?

The innovation that I see in this paper is very specific, the use of stable isotopes to very specific types of waters. The uniqueness of the specific case is the specific O-H numbers that create very distinct end members, that possibly can lead to the ability to distinguish the mixture at very wide range of mixing ratios (this however is not explored). But mixtures of two end members, including by O-H isotopes, is not new, as the authors acknowledge

Therefore, my requirement for this paper to be accepted is to improve the description of the unique case study throughout the manuscript, so that the innovation will be much better explained, highlighted, and discussed. If properly done this can probably be a good reference study

On a more technical level: 1. Since the authors describe model predictions, I'm not sure if past tense is the appropriate. In the same context, instead of "results show" I'd use "results suggest"

2. I do not think the model neglects fractionation (line 118). It is you (the authors) that neglect such a process. But why neglect? Is there a reason to think such a process is relevant here?

---

## Author Comment (AC1) · 12 Aug 2018

We would like to thank the referee Prof. Alex Furman for his review and comments. We provide below our detailed response to each comment.

General comment: Sincerely, I am not sure how to treat this manuscript. On one hand addresses a question of increasing concern (globally, not only in Israel), and it describes a work properly conducted, including a very nice combination of isotopic work and a flow and transport model. On the other hand, I'm asking myself at the end of the

[Figure]

day "what's new?" Is there something that can really be learned from this manuscript? Is it describing something that was not clear from the beginning? The innovation that I see in this paper is very specific, the use of stable isotopes to very specific types of waters. The uniqueness of the specific case is the specific O-H numbers that create very distinct end members, that possibly can lead to the ability to distinguish the mixture at very wide range of mixing ratios (this however is not explored). But mixtures of two end members, including by O-H isotopes, is not new, as the authors acknowledge. Therefore, my requirement for this paper to be accepted is to improve the description of the unique case study throughout the manuscript, so that the innovation will be much better explained, highlighted, and discussed. If properly done this can probably be a good reference study.

Response to general comment: We will emphasize the uniqueness of this work in the revised manuscript as detailed below:

1) The uniqueness of this case-study and its importance The uniqueness of this study, as the reviewer stated, is based on the specific case where reverse-osmosis desalinated seawater (DSW) are being recharged into natural fresh groundwater (GW). There are currently only a few places that are practicing managed aquifer recharge (MAR) with DSW, but this practice will grow due to the increasing use of DSW globally. Practically all the known case studies of MAR with DSW involve brackish-water aquifers (in the Gulf countries) and not necessarily reverse-osmosis DSW. These two features are important for the contrast in heavy isotopes concentrations to be large enough to enable simple mixing calculations.

2) The novel use of isotope-transport modeling The advantage of using stable water isotopes to trace DSW in the SAT site of the Shafdan MAR system, Israel, was recently reported by Negev et al. (2017). However, to the best of our knowledge, this is the first paper reporting groundwater modeling of reverse-osmosis desalinated seawater using isotope data. Combining a calibrated flow and transport model with isotope data enables prediction of the DSW plume spreading in the aquifer in future scenarios. This

is a quantitative step ahead of previous isotope-analysis studies related to reverse-osmosis DSW in aquifers.

3) The robustness of the two end-member assumption and its validation In this work we use two features of stable water isotopes that are mutually related, in order to trace reverse-osmosis DSW in the aquifer: (1) the distinct difference between isotope composition of reverse-osmosis DSW and natural fresh water; and (2) the relative similarity of isotope composition among natural fresh water, which simplify the system to a binary mixture. See section 3.1, lines 139-140 and 151-154. The binary simplification is a robust tool for modeling DSW spreading in the aquifer, as it reduces considerably the amount of isotope data that needs to be collected and analyzed for boundary conditions. Other modeling studies that use water isotopes, implement detailed transient boundary conditions based on large data sets that vary spatially and temporally (e.g., Reynolds and Marimuthu, 2007; Liu et al., 2014) that in most cases are not available. Mixing of two end-members is indeed widespread in geochemical studies, and can also be found in few groundwater modeling studies that use stable water isotopes as tracers (e.g., Krabbenhoft et al., 1990; Stichler et al., 2008). However, in this work an original error analysis is used to estimate the validity of the two-end-members assumption (section 3.2.3). This error analysis is possible due to the unique use of the flow and transport models discussed previously. We showed in the analysis (Figure 6) that the assumption is valid for the current isotope-composition variability that is found in the aquifer ($\delta$18O=–4.48 to –5.43‰ and $\delta$2H=–18.41 to –22.68‰. We also note in the original manuscript the conditions that permit the use of a two end-members approach (lines 215-218).

All the above clarifications will be inserted to the revised manuscript to describe more explicitly the novelty and uniqueness of this work.

Comment 1: Since the authors describe model predictions, I'm not sure if past tense is the appropriate. In the same context, instead of "results show" I'd use "results suggest"

Response 1: We agree. We will revise the manuscript according to the reviewer's suggestion.

Comment 2: I do not think the model neglects fractionation (line 118). It is you (the authors) that neglect such a process. But why neglect? Is there a reason to think such a process is relevant here?

Response 2: Isotope fractionation processes such as evaporation or biogeochemical interactions are negligible in the transport of water-isotopes in deep groundwater (depth of water table » depth of root zone). There is strong evidence that local groundwater tends to be isotopically uniform (e.g., Krabbenhoft et al., 1990 and reference therein). Moreover, our measurements at this fast-percolating MAR site show similar isotope composition between the DSW source-water at the surface, in the variably-saturated zone and at the shallow GW (Ganot et al., 2018). Therefore, isotope fractionation even if exists in the aquifer to some extent as a slow process (centuries-millennia), it should defiantly consider negligible compared to the distinct difference between the isotope end-members. Note that the error analysis presented in section 3.2.3 (and discussed here previously) aimed to test the binary end-members assumption, justify the assumption that isotope fractionation is negligible in this GW system.

We will add this explanation to the revised manuscript (in line 118 and also in section 3.2.3).

References

Ganot, Y., Holtzman, R., Weisbrod, N., Russak, A., Katz, Y. and Kurtzman, D.: Geochemical Processes During Managed Aquifer Recharge With Desalinated Seawater, Water Resources Research, 54(2), 978–994, doi:10.1002/2017WR021798, 2018.

Krabbenhoft, D. P., Anderson, M. P. and Bowser, C. J.: Estimating groundwater exchange with lakes: 2. Calibration of a three-dimensional, solute transport model to a stable isotope plume, Water Resour. Res., 26(10), 2455–2462,
doi:10.1029/WR026i010p02455, 1990.

Negev, I., Guttman, J. and Kloppmann, W.: The Use of Stable Water Isotopes as Tracers in Soil Aquifer Treatment (SAT) and in Regional Water Systems, Water, 9(2), 73, doi:10.3390/w9020073, 2017.

Liu, Y., Yamanaka, T., Zhou, X., Tian, F. and Ma, W.: Combined use of tracer approach and numerical simulation to estimate groundwater recharge in an alluvial aquifer system: A case study of Nasunogahara area, central Japan, Journal of Hydrology, 519, 833–847, doi:10.1016/j.jhydrol.2014.08.017, 2014.

Reynolds, D. A. and Marimuthu, S.: Deuterium composition and flow path analysis as additional calibration targets to calibrate groundwater flow simulation in a coastal wetlands system, Hydrogeol J, 15(3), 515–535, doi:10.1007/s10040-006-0113-5, 2007.

Stichler, W., Maloszewski, P., Bertleff, B. and Watzel, R.: Use of environmental isotopes to define the capture zone of a drinking water supply situated near a dredge lake, Journal of Hydrology, 362(3), 220–233, doi:10.1016/j.jhydrol.2008.08.024, 2008.

---

## Referee Comment (RC2) · Anonymous Referee #2 · 11 Oct 2018

As an increased using of new water resources, understanding its effect is necessary. The desalinated seawater is a potential water resource with the increasing demands of fresh water all around the world in the modern days. This manuscript introduces a new insight of stable isotope application in water research. An isotopic solute transport model was built to estimate the spread of the desalinated seawater in the coastal aquifer. It is not common to see applying the stable isotope method by a numerical model, considering the wide range of isotopes of the water bodies. Authors take the advantage of the conservation isotope concentration of the DSW, so that they can sim-

plify the local water sources to groundwater into a binary system in the sight of isotopic concentration. To be honest, this manuscript is very interesting but it is really hard for me to make a comment on it. Generally, it is a useful and convincing study, but it lacks novelty on method. So my suggestion is accepted after revision.

The number and type of water samples are not clear in this manuscript. Since the rainwater and the runoff water in the setting pond are regarded as water sources to the groundwater, it should be more specific of the isotope data, especially, the runoff water.

Q1: How many isotopic water samples do you have? Q2: Is there any water samples of rain or runoff water in the ponds? Line 124 Q3: How did you deal with these values? The lowest and highest? For example, the lowest CGW $\delta 18O=-5.43‰$ $\delta 2H=-22.68‰$ is this the result of one sample or the lowest values of all groundwater samples? Why? Line 121. Q4: Is there any soluble salt in the study area? Line 150 Q5: Do you have any data from modern rainwater? Is that too old for your study? 1972? Line 153. All the advantages listed in the manuscript of using the method show that it is specific lucky for this area. Can you give some descriptions to show the universal applicability?

---

## Author Comment (AC2) · 19 Oct 2018

We would like to thank anonymous referee #2 for his positive review and insightful comments. We provide below our detailed response to each comment.

**Referee overview** As an increased using of new water resources, understanding its effect is necessary. The desalinated seawater is a potential water resource with the increasing demands of fresh water all around the world in the modern days. This manuscript introduces a new insight of stable isotope application in water research.

[Figure]

An isotopic solute transport model was built to estimate the spread of the desalinated seawater in the coastal aquifer. It is not common to see applying the stable isotope method by a numerical model, considering the wide range of isotopes of the water bodies. Authors take the advantage of the conservation isotope concentration of the DSW, so that they can simplify the local water sources to groundwater into a binary system in the sight of isotopic concentration. To be honest, this manuscript is very interesting but it is really hard for me to make a comment on it. Generally, it is a useful and convincing study, but it lacks novelty on method. So my suggestion is accepted after revision.

**Response to the Referee overview** We are very contented that referee #2 summarized the significance of this work in its two most unique features. The extreme difference in the isotopic signature between reverse-osmosis DSW and all other sources of natural fresh water that makes the binary system approximation solid, and the database flow and transport model of water isotopes and its simulations. The novelty of this study was discussed in our response to referee #1 (https://doi.org/10.5194/hess-2018-341-AC1).

**General comment** The number and type of water samples are not clear in this manuscript. Since the rainwater and the runoff water in the setting pond are regarded as water sources to the groundwater, it should be more specific of the isotope data, especially, the runoff water.

**Response to General comment** We will add a detailed description of the water samples to the revised manuscript, including water sample type and number of samples. Please see more details in the following responses to the specific comments.

**Comment 1** How many isotopic water samples do you have?

**Response 1** We have a total of 70 isotope samples, where each sample was analyzed for $^{18}O$ and $^2H$ isotopes. We used 42 samples that were sampled from groundwater production wells for the model calibration (see Fig. 4c). The remaining 28 samples
were sampled from several sources: (1) the DSW inlet pipe, (2) few locations inside the DSW infiltration pond, (3) runoff canal, and (4) shallow observation wells (OA and OB). We will add this information to the revised manuscript.

**Comment 2** Is there any water samples of rain or runoff water in the ponds? Line 124

**Response 2** We did not sample rainwater in this study and instead we used the rainwater data from Gat and Dansgaard (1972) as described in lines 152-154. We have water samples from one runoff event ($\delta^{18}O$=–4.77‰ and $\delta^2H$=–19.5‰ ) that was taken on January 2017 (very similar to the rain composition of $\delta2H$=–19.9‰ line 154). Unfortunately, we have no samples from the runoff event of 2015 (line135-136) and we assumed the runoff isotope composition of 2015 is similar to 2017. This clarification will be added to the revised manuscript.

**Comment 3** How did you deal with these values? The lowest and highest? For example, the lowest $C_{GW}$ $\delta^{18}O$=–5.43‰ $\delta^2H$=–22.68‰ is this the result of one sample or the lowest values of all groundwater samples? Why? Line 121.

**Response 3** We normalized the concentrations to the lowest value that was measured in the aquifer (i.e., based on one sample). Next, in section 3.2.3 (line 205) we check this assumption by normalizing the concentrations to the highest value (also based on one sample) and then compare the simulation results of these two extreme options (Fig. 6). We show that the results are almost similar (less than 1‰ difference) in the aquifer area next to the infiltration ponds. Hence, in similar isotope binary systems, a practical conclusion of this analysis will be to use an average isotope value to normalize the isotope concentrations in the aquifer. We will add this explanation to the revised manuscript.

**Comment 4** Is there any soluble salt in the study area? Line 150

**Response 4** There is no extensive soluble salt layer/formation according to the recent available geological data. In Ganot et al., 2018 (Table 1) it is shown that chloride concentrations are similar in: (1) pond water of the infiltration basin; (2) variably saturated zone (suction cups at 0.5-3 m below surface); and (3) shallow groundwater below the pond. Therefore, dissolution of chloride during infiltration is negligible. Chloride concentration of naturally infiltrating groundwater is mostly related to evapotranspiration which concentrates the rainwater with respect to the conservative chloride ion. Nevertheless, the binary system approach used in this study based on conservative water-isotope tracers is superior to both conservative and non-conservative classic tracers. This is an important insight - thank you. We will add this paragraph to the revised manuscript.

**Comment 5** Do you have any data from modern rainwater? Is that too old for your study? 1972? Line 153

**Response 5** The paper of Gat and Dansgaard (1972) is the most comprehensive survey published on stable isotopes of fresh water in Israel. It includes rain samples that were collected next to the Menashe MAR site and were used in this study (line 153). A more recent study (Goldsmith et al., 2017) reports similar values ($\delta^{18}$O=–5.1‰ to –5.7; $\delta^2$H=–18.6‰ to =–25.6‰ ) for rain samples collected at the coastal plain of Israel, but it based on fewer sampling station and with no sampling point next to Menashe MAR site. We will add this reference to the manuscript.

**Comment 6** All the advantages listed in the manuscript of using the method show that it is specific lucky for this area. Can you give some descriptions to show the universal applicability?

**Response 6** The advantage of using stable water isotopes for tracing reverse-osmosis desalinated water in various downstream water systems is already known from previous studies (Ganot et al., 2018; Kloppmann et al., 2008a, 2008b; Kloppmann et al., 2018; Negev et al., 2017). In this study we use this advantage in a modeling framework to predict future mixing and spreading trends of DSW in the aquifer (line 56-63). Therefore, the modeling approach presented in this study can be used in other sites

(e.g., Mazariegos et al., 2017; Negev et al., 2017; Stuyfzand et al., 2017) to predict reverse-osmosis desalinated-water distribution in aquifers. As the production of DSW using reverse-osmosis is projected to increase (Hanasaki et al., 2016) and the use of MAR system widens (Hartog and Stuyfzand, 2017; Rodríguez-Escales et al., 2018) we are sure that the advantage listed in this manuscript is/will be highly relevant for more MAR hydrologists. Above seawater desalination and MAR, integrating water resources is a key for dealing with increasing water demands and droughts, hence operations of similar features are expected to develop first in semi-arid regions but also in more temperate climates. We will add these arguments to the revised manuscript.

**References**

Ganot, Y., Holtzman, R., Weisbrod, N., Russak, A., Katz, Y. and Kurtzman, D.: Geochemical Processes During Managed Aquifer Recharge With Desalinated Seawater, Water Resources Research, 54(2), 978–994, doi:10.1002/2017WR021798, 2018.

Gat, J. R. and Dansgaard, W.: Stable isotope survey of the fresh water occurrences in Israel and the Northern Jordan Rift Valley, Journal of Hydrology, 16(3), 177–211, doi:10.1016/0022-1694(72)90052-2, 1972.

Goldsmith, Y., Polissar, P. J., Ayalon, A., Bar-Matthews, M., deMenocal, P. B. and Broecker, W. S.: The modern and Last Glacial Maximum hydrological cycles of the Eastern Mediterranean and the Levant from a water isotope perspective, Earth and Planetary Science Letters, 457, 302–312, doi:10.1016/j.epsl.2016.10.017, 2017.

Hanasaki, N., Yoshikawa, S., Kakinuma, K. and Kanae, S.: A seawater desalination scheme for global hydrological models, Hydrol. Earth Syst. Sci., 20(10), 4143–4157, doi:10.5194/hess-20-4143-2016, 2016.

Hartog, N., Stuyfzand, P., Hartog, N. and Stuyfzand, P. J.: Water Quality Considerations on the Rise as the Use of Managed Aquifer Recharge Systems Widens, Water, 9(10), 808, doi:10.3390/w9100808, 2017.

Kloppmann, W., Vengosh, A., Guerrot, C., Millot, R. and Pankratov, I.: Isotope and Ion Selectivity in Reverse Osmosis Desalination: Geochemical Tracers for Man-made Freshwater, Environmental Science Technology, 42(13), 4723–4731, doi:10.1021/es7028894, 2008a.

Kloppmann, W., Van Houtte, E., Picot, G., Vandenbohede, A., Lebbe, L., Guerrot, C., Millot, R., Gaus, I. and Wintgens, T.: Monitoring Reverse Osmosis Treated Wastewater Recharge into a Coastal Aquifer by Environmental Isotopes (B, Li, O, H), Environmental Science Technology, 42(23), 8759–8765, doi:10.1021/es8011222, 2008b.

Kloppmann, W., Negev, I., Guttman, J., Goren, O., Gavrieli, I., Guerrot, C., Flehoc, C., Pettenati, M. and Burg, A.: Massive arrival of desalinated seawater in a regional urban water cycle: A multi-isotope study (B, S, O, H), Science of The Total Environment, 619–620, 272–280, doi:10.1016/j.scitotenv.2017.10.181, 2018.

Mazariegos, J. G., Walker, J. C., Xu, X. and Czimczik, C. I.: Tracing Artificially Recharged Groundwater using Water and Carbon Isotopes, Radiocarbon; Tucson, 59(2), 407–421, doi:http://dx.doi.org/10.1017/RDC.2016.51, 2017.

Negev, I., Guttman, J. and Kloppmann, W.: The Use of Stable Water Isotopes as Tracers in Soil Aquifer Treatment (SAT) and in Regional Water Systems, Water, 9(2), 73, doi:10.3390/w9020073, 2017.

Rodríguez-Escales, P., Canelles, A., Sanchez-Vila, X., Folch, A., Kurtzman, D., Rossetto, R., Fernández-Escalante, E., Lobo-Ferreira, J.-P., Sapiano, M., San-Sebastián, J. and Schüth, C.: A risk assessment methodology to evaluate the risk Failure of managed aquifer recharge in the Mediterranean Basin, Hydrology and Earth System Sciences, 22(6), 3213–3227, doi:10.5194/hess-22-3213-2018, 2018.

Stuyfzand, P. J., Smidt, E., Zuurbier, K. G., Hartog, N. and Dawoud, M. A.: Observations and Prediction of Recovered Quality of Desalinated Seawater in the Strategic ASR Project in Liwa, Abu Dhabi, Water, 9(3), 177, doi:10.3390/w9030177, 2017.

---

## Author Response (AR1)

We would like to thank the Editor for handling the manuscript and to thank the Referees for their insightful comments, which have helped to improve the manuscript. We provide below our detailed response to each comment (in blue). All line numbers refer to the revised marked manuscript that follows the comments and responses (insertions marked in blue and deletions in red strikethrough).

**Referee #1**

This manuscript describes a case where water of unique quality recharges a local aquifer. The manuscript looks experimentally and through MODFLOW modeling at the expansion and mixing of the new water in the aquifer for a specific setup where recovery wells are located around the recharge pond. The concern here is desalinated water, and the main tool to distinguish these water from the background is stable isotopes.

We would like to thank the referee Prof. Alex Furman for his review and comments. We provide below our detailed response to each comment.

General comment: Sincerely, I am not sure how to treat this manuscript. On one hand addresses a question of increasing concern (globally, not only in Israel), and it describes a work properly conducted, including a very nice combination of isotopic work and a flow and transport model. On the other hand, I'm asking myself at the end of the day "what's new?" Is there something that can really be learned from this manuscript? Is it describing something that was not clear from the beginning? The innovation that I see in this paper is very specific, the use of stable isotopes to very specific types of waters. The uniqueness of the specific case is the specific O-H numbers that create very distinct end members, that possibly can lead to the ability to distinguish the mixture at very wide range of mixing ratios (this however is not explored). But mixtures of two end members, including by O-H isotopes, is not new, as the authors acknowledge. Therefore, my requirement for this paper to be accepted is to improve the description of

the unique case study throughout the manuscript, so that the innovation will be much better explained, highlighted, and discussed. If properly done this can probably be a good reference study.

Response to general comment: We emphasized the uniqueness of this work in the revised manuscript as detailed below:

1) The uniqueness of this case-study and its importance: The uniqueness of this study, as the reviewer stated, is based on the specific case where reverse-osmosis desalinated seawater (DSW) are being recharged into natural fresh groundwater (GW). There are currently only a few places that are practicing managed aquifer recharge (MAR) with DSW, but this practice will grow due to the increasing use of DSW globally. Practically all the known case studies of MAR with DSW involve brackish-water aquifers (in the Gulf countries) and not necessarily reverse-osmosis DSW. These two features are important for the contrast in heavy isotopes concentrations to be large enough to enable simple mixing calculations. (lines 56-61).

2) The novel use of isotope-transport modeling: The advantage of using stable water isotopes to trace DSW in the SAT site of the Shafdan MAR system, Israel, was recently reported by Negev et al. (2017). However, to the best of our knowledge, this is the first paper reporting groundwater modeling of reverse-osmosis desalinated seawater using isotope data. Combining a calibrated flow and transport model with isotope data enables prediction of the DSW plume spreading in the aquifer in future scenarios. This is a quantitative step ahead of previous isotope-analysis studies related to reverse-osmosis DSW in aquifers (lines 65-67).

3) The robustness of the two end-member assumption and its validation: In this work we use two features of stable water isotopes that are mutually related, in order to trace reverse-osmosis DSW in the aquifer: (1) the distinct difference between isotope composition of reverse-osmosis DSW and natural fresh water; and (2) the relative similarity of isotope composition among natural fresh water, which simplify the system to a binary mixture. See section 3.1, lines 146-147 and 160-165. The binary simplification is a robust tool for modeling DSW spreading in the aquifer, as it reduces considerably the amount of isotope data that needs to be collected and analyzed for boundary conditions. Other modeling studies that use water isotopes, implement detailed transient boundary conditions based on large data sets that vary

spatially and temporally (e.g., Reynolds and Marimuthu, 2007; Liu et al., 2014) that in most cases are not available. Mixing of two end-members is indeed widespread in geochemical studies, and can also be found in few groundwater modeling studies that use stable water isotopes as tracers (e.g., Krabbenhoft et al., 1990; Stichler et al., 2008). However, in this work an original error analysis is used to estimate the validity of the two-end-members assumption (section 3.2.3). This error analysis is possible due to the unique use of the flow and transport models discussed previously. We showed in the analysis (Fig. 6) that the assumption is valid for the current isotope-composition variability that is found in the aquifer ( $\delta^{18}O=-4.48$  to -5.43% and  $\delta^{2}H=-18.41$  to -22.68%). We also note in the original manuscript the conditions that permit the use of a two end-members approach (lines 226-229).

Comment 1: Since the authors describe model predictions, I'm not sure if past tense is the appropriate. In the same context, instead of "results show" I'd use "results suggest".

Response 1: We agree. We revised the manuscript according to the reviewer's suggestion.

Comment 2: I do not think the model neglects fractionation (line 118). It is you (the authors) that neglect such a process. But why neglect? Is there a reason to think such a process is relevant here?

Response 2: Isotope fractionation processes such as evaporation or biogeochemical interactions are negligible during the transport of water-isotopes in deep groundwater (depth of water table >> depth of root zone). There is strong evidence that local groundwater tends to be isotopically uniform (e.g., Krabbenhoft et al., 1990 and reference therein). Moreover, our measurements at this fast-percolating MAR site show similar isotope composition between the DSW source-water at the surface, in the variably-saturated zone and at the shallow GW (Ganot et al., 2018). Therefore, isotope fractionation even if exists in the aquifer to some extent as a slow process (centuries-millennia), it should defiantly consider negligible compared to the distinct difference between the isotope end-members. Note that the error analysis presented in section 3.2.3 (and discussed here previously) aimed to test the binary end-members

assumption, justify the assumption that isotope fractionation is negligible in this GW system. We added this explanation to the revised manuscript (in lines 125-126 and also in section 3.2.3, lines 235-240).

**Referee #2**

As an increased using of new water resources, understanding its effect is necessary. The desalinated seawater is a potential water resource with the increasing demands of fresh water all around the world in the modern days. This manuscript introduces a new insight of stable isotope application in water research. An isotopic solute transport model was built to estimate the spread of the desalinated seawater in the coastal aquifer. It is not common to see applying the stable isotope method by a numerical model, considering the wide range of isotopes of the water bodies. Authors take the advantage of the conservation isotope concentration of the DSW, so that they can simplify the local water sources to groundwater into a binary system in the sight of isotopic concentration. To be honest, this manuscript is very interesting but it is really hard for me to make a comment on it. Generally, it is a useful and convincing study, but it lacks novelty on method. So my suggestion is accepted after revision.

We are very contented that referee #2 summarized the significance of this work in its two most unique features. The extreme difference in the isotopic signature between reverse-osmosis DSW and all other sources of natural fresh water that makes the binary system approximation solid, and the data-based flow and transport model of water isotopes and its simulations. The novelty of this study was discussed in our response to the general comment of referee #1.

General comment: The number and type of water samples are not clear in this manuscript. Since the rainwater and the runoff water in the setting pond are regarded as water sources to the groundwater, it should be more specific of the isotope data, especially, the runoff water.

Response to General comment: We added a detailed description of the water samples to the revised manuscript, including water sample type and number of samples (lines 86-88 and Table S1 in the Supplement). Please see more details in the following responses to the specific comments.

Comment 1: How many isotopic water samples do you have?

Response 1: We have a total of 61 isotope samples, where each sample was analyzed for 18O and 2H isotopes. 42 samples were sampled from groundwater production wells. The remaining samples were sampled from several sources: (1) the DSW inlet pipe, (2) few locations inside the DSW infiltration pond, (3) runoff canal, and (4) shallow observation wells (OA and OB). We added this information to the revised manuscript (lines 86-88 and Table S1 in the Supplement).

**Comment 2: Is there any water samples of rain or runoff water in the ponds? Line 124**

Response 2: We did not sample rainwater in this study and instead we used the rain-water data from Gat and Dansgaard (1972) as described in lines 162-163. We have water samples from one runoff event ( $\delta^{18}O=-4.77\%$  and  $\delta^{2}H=-19.5\%$ ) that was taken on January 2017 (very similar to the rain composition of  $\delta^{2}H=-19.9\%$  line 162). Unfortunately, we have no samples from the runoff event of 2015 (line 142-143) and we assumed the runoff isotope composition of 2015 is similar to 2017. This information added to the revised manuscript (lines 86-88 and Table S1 in the Supplement).

Comment 3: How did you deal with these values? The lowest and highest? For example, the lowest  $C_{GW}$  $\delta^{18}O=-5.43\%$   $\delta^{2}H=-22.68\%$  is this the result of one sample or the lowest values of all groundwater samples? Why? Line 121.

Response 3: We normalized the concentrations to the lowest value that was measured in the aquifer (i.e., based on one sample). Next, in section 3.2.3 (line 215) we check this assumption by normalizing the concentrations to the highest value (also based on one sample) and then compare the simulation results of these two extreme options (Fig. 6). We show that the results are almost similar (less than 1‰ difference) in the aquifer area next to the infiltration ponds. Hence, in similar isotope binary systems, a

practical conclusion of this analysis will be to use an average isotope value to normalize the isotope concentrations in the aquifer. We added this explanation to the revised manuscript (lines 229-232)

**Comment 4: Is there any soluble salt in the study area? Line 150**

Response 4: There is no extensive soluble salt layer/formation according to the recent available geological data. In Ganot et al., 2018 (Table 1) it is shown that chloride concentrations are similar in: (1) pond water of the infiltration basin; (2) variably saturated zone (suction cups at 0.5-3 m below surface); and (3) shallow groundwater below the pond. Therefore, dissolution of chloride during infiltration is negligible. Chloride concentration of naturally infiltrating groundwater is mostly related to evapotranspiration which concentrates the rainwater with respect to the conservative chloride ion. Nevertheless, the binary system approach used in this study based on conservative water-isotope tracers is superior to both conservative and nonconservative classic tracers. This is an important insight - thank you. We added this information to the revised manuscript (lines 155-156 and 163-165).

Comment 5: Do you have any data from modern rainwater? Is that too old for your study? 1972? Line 153

Response 5: The paper of Gat and Dansgaard (1972) is the most comprehensive survey published on stable isotopes of fresh water in Israel. It includes rain samples that were collected next to the Menashe MAR site and were used in this study. A more recent study (Goldsmith et al., 2017) reports similar values  $(\delta^{18}O=-5.1\% \text{ to } -5.7; \delta^{2}H=-18.6\% \text{ to } =-25.6\%)$  for rain samples collected at the coastal plain of Israel, but it based on fewer sampling station and with no sampling point next to Menashe MAR site. We added this reference to the manuscript (line 163).

Comment 6: All the advantages listed in the manuscript of using the method show that it is specific lucky for this area. Can you give some descriptions to show the universal applicability?

Response 6: The advantage of using stable water isotopes for tracing reverse-osmosis desalinated water in various downstream water systems is already known from previous studies (Ganot et al., 2018; Kloppmann et al., 2008a, 2008b; Kloppmann et al.,2018; Negev et al., 2017). In this study we use this advantage in a modeling framework to predict future mixing and spreading trends of DSW in the aquifer. Therefore, the modeling approach presented in this study can be used in other sites (e.g., Mazariegos et al., 2017; Negev et al., 2017; Stuyfzand et al., 2017) to predict reverse-osmosis desalinated-water distribution in aquifers. As the production of DSW using reverse-osmosis is projected to increase (Hanasaki et al., 2016) and the use of MAR system widens (e.g., Rodríguez-Escales et al., 2018) we are sure that the advantage listed in this manuscript is/will be highly relevant for more MAR hydrologists. Above seawater desalination and MAR, integrating water resources is a key for dealing with increasing water demands and droughts, hence operations of similar features are expected to develop first in semiarid regions but also in more temperate climates. We added these arguments to the revised manuscript (lines 56-61 and 272-277).

 and Daniel Kurtzman1

[revised manuscript text omitted]